# Light Emission from Fe^2+^-EGTA-H_2_O_2_ System Depends on the pH of the Reaction Milieu within the Range That May Occur in Cells of the Human Body

**DOI:** 10.3390/molecules29174014

**Published:** 2024-08-25

**Authors:** Krzysztof Sasak, Michal Nowak, Anna Wlodarczyk, Agata Sarniak, Wieslaw Tryniszewski, Dariusz Nowak

**Affiliations:** 1Department of Medical Imaging Techniques, Medical University of Lodz, Lindleya 6, 90-131 Lodz, Poland; krzysztof.sasak@umed.lodz.pl; 2Radiation Protection, University Hospital No. 2, Medical University of Lodz, Zeromskiego 113, 90-549 Lodz, Poland; m.nowak@skwam.lodz.pl; 3Department of Sleep Medicine and Metabolic Disorders, Medical University of Lodz, Mazowiecka 6/8, 92-215 Lodz, Poland; anna.wlodarczyk@umed.lodz.pl; 4Department of Clinical Physiology, Medical University of Lodz, Mazowiecka 6/8, 92-215 Lodz, Poland; agata.sarniak@umed.lodz.pl; 5Department of Radiological and Isotopic Diagnostics and Therapy, Medical University of Lodz, Zeromskiego 113, 90-549 Lodz, Poland; wieslaw.tryniszewski@umed.lodz.pl

**Keywords:** chemiluminescence, Fenton system, hydroxyl radicals, pH of reaction milieu, singlet oxygen

## Abstract

A Fe^2+^-EGTA(ethylene glycol-bis (β-aminoethyl ether)-*N*,*N*,*N*′,*N*′-tetraacetic acid)-H_2_O_2_ system emits photons, and quenching this chemiluminescence can be used for determination of anti-hydroxyl radical (•OH) activity of various compounds. The generation of •OH and light emission due to oxidative damage to EGTA may depend on the buffer and pH of the reaction milieu. In this study, we evaluated the effect of pH from 6.0 to 7.4 (that may occur in human cells) stabilized with 10 mM phosphate buffer (main intracellular buffer) on a chemiluminescence signal and the ratio of this signal to noise (light emission from medium alone). The highest signal (4698 ± 583 RLU) and signal-to-noise ratio (9.7 ± 1.5) were noted for pH 6.6. Lower and higher pH caused suppression of these variables to 2696 ± 292 RLU, 4.0 ± 0.8 at pH 6.2 and to 3946 ± 558 RLU, 5.0 ± 1.5 at pH 7.4, respectively. The following processes may explain these observations: enhancement and inhibition of •OH production in lower and higher pH; formation of insoluble Fe(OH)_3_ at neutral and alkaline environments; augmentation of •OH production by phosphates at weakly acidic and neutral environments; and decreased regeneration of Fe^2+^-EGTA in an acidic environment. Fe^2+^-EGTA-H_2_O_2_ system in 10 mM phosphate buffer pH 6.6 seems optimal for the determination of anti-•OH activity.

## 1. Introduction

### 1.1. Fenton Systems as a Tool for Determination of Anti-(•OH) Activity and Limiting Buffers Interactions

Hydroxyl radicals (•OH) are very reactive and their excessive formation in the human body is involved in numerous pathological processes [1] including rheumatoid arthritis [2], neurodegenerative disorders [3], cancer [4], and atherosclerosis [4,5]. Fenton and Fenton-like reactions are the most important sources of •OH in the human body [4]; however, other processes such as the enhanced formation of peroxynitrite with its subsequent decomposition can contribute to tissue •OH overload [6]. Apart from breaking chemical reactions between H_2_O_2_ and transition metal ions (mainly Fe^2+^, Fe^3+^ Cu^2+^, Cu^+^) with the specific metal chelators, application of drugs or dietary supplements that can directly react with •OH (•OH scavengers) before their oxidative attack on cellular biomolecules seems to be a possible preventive or therapeutic approach [7]. Compounds being possible candidates for such treatment should be tested in detail in vitro before experiments on laboratory animals and clinical trials. Aqueous systems based on Fenton or Fenton-like reactions are frequently used for in vitro evaluation of the antioxidant activity. Because •OH generation can be significantly influenced by the pH of the reaction milieu [8,9], it is necessary to keep stable hydronium ions (H_3_O^+^) activity during the experiment. Changes of pH related, for instance, to the addition of a tested compound may inhibit or enhance •OH generation and lead to false results. Unfortunately, the vast majority of buffering compounds can react with •OH and, in consequence, decrease the signal-to-noise ratio and repeatability of results. For instance, organic buffers such as Tris, Tricine, and Hepes were reported to effectively scavenge •OH radicals [10]. Even bicarbonate and phosphate buffers can react with •OH radicals [11,12,13] and may inhibit •OH reactions with an appropriate probe. It should be pointed out that Fenton’s systems dedicated to the evaluation of anti-•OH activity must resemble in vivo conditions and be relatively simple to facilitate the interpretation of obtained results. Bicarbonate and phosphate buffers are the main buffers in extracellular (e.g., blood) and intracellular fluid [14], respectively. There are also other intracellular buffers such as amino-acids, proteins, and organic acids (mainly carboxylic acids) (e.g., acetic acid, lactic acid, citric acid, succinic acid) that, with their dissociated form, can stabilize pH inside the cell [14]. On the other hand, •OH radicals can effectively oxidize the aforementioned intracellular buffers [15,16]. Moreover, oxidation of some proteins and tryptophane can lead to the formation of excited carbonyl groups with a subsequent photon emission [17,18,19], making results obtained with the Fe^2+^-EGTA-H_2_O_2_ system almost impossible to interpret. Therefore, phosphate buffer composed of dibasic sodium phosphate and monobasic sodium phosphate seems to be the optimal intracellular buffer that can be used for investigation of the effect of pH changes on UPE of Fe^2+^-EGTA-H_2_O_2_ system.

### 1.2. Properties of Fe^2+^-EGTA-H_2_O_2_ System Generating Ultra-Weak Chemiluminescence

We developed a system composed of Fe^2+^, EGTA (ethylene glycol-bis (β-aminoethyl ether)-*N*,*N*,*N*′,*N*′-tetraacetic acid), and H_2_O_2_, which generates •OH radicals (Fenton reaction) [20]. •OH radicals can attack and cleave the ether bond in an EGTA backbone structure, leading to the formation of products containing triplet excited carbonyl groups and subsequent ultra-weak photon emissions (UPE) [20]. Measurement of photon emanation (UPE within a defined time) with a sensitive luminometer could be a measure of •OH radicals production [20]. By using this system, we were able to determine pro-oxidant (enhancing •OH production) or anti-oxidant (inhibiting •OH activity) properties of various plant polyphenols at concentrations within the range 5 µmol/L to 50 µmol/L [21] and ascorbic acid [22]. These experiments were performed in phosphate-buffered saline pH = 7.4 [20,21,22] containing 137 mmol/L NaCl, 2.7 mmol/L KCl, and 10 mmol/L phosphate. This relatively high concentration of Cl- can suppress •OH radicals’ activity before their reaction with EGTA, leading to a decrease in photon emanations and low chemiluminescence signal. Apart from the triplet excited carbonyl groups, the decay of singled oxygen (O_2_ (1Δg)) that is formed by a Fenton reagent (Fe^2+^, H_2_O_2_) could be a source of emitted photons from the Fe^2+^-EGTA-H_2_O_2_ system [14,23]. These photons have three characteristic bands of emission at 1270 nm, 703 nm, and 634 nm [20]. The spectral range of the luminometer used in our experiments was from 380 nm to 630 nm. Therefore, it cannot be ruled out that wavelength photons of 634 nm may contribute to some extent to UPE of Fe^2+^-EGTA-H_2_O_2_ system. Although we excluded indirectly the significant contribution of these photons to UPE of Fe^2+^-EGTA-H_2_O_2_ system with the use of sodium azide an O_2_ (1Δg) scavenger [20], it would be better to complete these experiments and analyze a UPE signal from a system that specifically generated O_2_ (1Δg). If the luminometer is sensitive to 634 nm photons, it will lead to over- or underestimation of •OH scavenging activity of the tested compound, depending on the simultaneous reactivity with O_2_ (1Δg).

### 1.3. Aims of the Study

To better characterize the Fe^2+^-EGTA-H_2_O_2_ system as a tool for measurement of anti-•OH activity, we therefore evaluated the effect of a medium composed of 10 mmol/L phosphate buffer of pH ranging from 6.0 to 7.4 on UPE, ΔUPE (increment in UPE calculated as the difference between UPE and noise), and the UPE signal-to-noise ratio (light emission from medium alone). Moreover, we analyzed the effect of the system generating O_2_ (1Δg) composed of H_2_O_2_ and sodium hypochlorite (NaOCl) on a UPE signal as well as O_2_ (1Δg) decay-dependent photon emission from Fe^2+^-H_2_O_2_ recorded by a luminometer with a photomultiplier spectrum from 380 nm to 630 nm.

We found that the decay of O_2_ (1Δg) did not significantly affect the UPE of the Fe^2+^-EGTA-H_2_O_2_ system and that pH = 6.6 of reaction milieu results in maximal signal-to-noise ratio being optimal for in vitro determination of anti-•OH activities of various compounds.

## 2. Results

### 2.1. Effect of Singlet Oxygen (O_2_ (1Δg)) Generating System (H_2_O_2_-NaOCl) on the Luminescence Signal Recorded by Luminometer with Photomultiplier Spectrum from 380 nm to 630 nm

The baseline signal (UPE of medium alone-PB pH = 6.8 with injected NaCl) was 609 ± 60 (630; 107) RLU. The addition of H_2_O_2_ or NaOCl alone did not change UPE 628 ± 74 (651; 120) RLU and 651 ± 80 (695; 133) RLU, respectively. Light emission from the O2 (1Δg) generating system (H_2_O_2_-NaOCl) reached 926 ± 245 (878; 218) RLU and the median value was 1.37-times higher than that of the baseline (Table 1). NaN_3_ did not suppress UPE of H_2_O_2_–NaOCl (1073 ± 105 (1102; 96) RLU). Similar results were obtained for the second series of experiments with PB pH = 6.6. H_2_O_2_–NaOCl increased median RLU 1.28 times (Table 1). However, surprisingly, light emission from H_2_O_2_-NaOCl-NaN_3_ was higher (*p* < 0.05) than that of H_2_O_2_-NaOCl.

### 2.2. Effect of pH of Reaction Milieu on Light Emission from Fe^2+^-EGTA-H_2_O_2_ System

Figure 1 shows the effect of pH of reaction milieu (changes from 6.0 to 7.4) on the ratios of UPE and ΔUPE to noise. They were highest (*p* < 0.05) at pH 6.6 and were 9.7 ± 1.5 (9.3; 2.3) and 8.7 ± 1.5 (8.3; 2.3) (Figure 1). The UPE of the Fe^2+^-H_2_O_2_ system was small and ranged between 620 ± 70 RLU (at pH 6.6) and 826 ± 76 RLU (at pH 7.2) as well as the ratios of UPE of the Fe^2+^-H_2_O_2_ system to baseline were low at the whole studied pH range and did not exceed 1.3 (Figure 1). Similarly, UPE and ΔUPE of the Fe^2+^-EGTA-H_2_O_2_ system revealed maximal values for pH 6.6 and 6.8 (*p* < 0.05) 4698 ± 583 RLU (4557; 1062), 4207 ± 586 RLU (4006; 1069) and 4651 ± 410 RLU (4756; 651), 4004 ± 387 RLU (4144; 605), respectively. For higher pH, they gradually decreased (*p* < 0.05) and reached 3946 ± 558 RLU (3745; 465) and 3023 ± 658 RLU (2983; 544) at pH = 7.4. Lowering pH (more acidic environment) resulted in suppression of UPE and ΔUPE to values of 2696 ± 292 RLU (2674; 345) and 2001 ± 340 (1991; 198) RLU at pH 6.2 (*p* < 0.05) (Appendix A). More details on UPE and ratios of UPE to noise of the Fe^2+^-EGTA-H_2_O_2_ system and controls are shown in Appendix A.

## 3. Discussion

### 3.1. Contribution of 634 nm Photons Derived from Singlet Oxygen Decay to Maximal Signal Generated by the Fe^2+^-EGTA-H_2_O_2_ System

Light emission from Fe^2+^-EGTA-H_2_O_2_ was tested in PB with pH from 6.0 to 7.4. The maximal signal and optimal signal-to-noise ratio were observed at pH 6.6 and 6.8. In the majority of cells, the intracellular pH is within the range of 6.7 to 7.2 or even lower in some cellular organelles such as lysosomes where it reached 4.5–5.0 [24]. Therefore, our findings are relevant to conditions inside the cells where H_2_O_2_ can leak from mitochondria and is involved in intracellular signaling [25] and also reacts with iron to form •OH radicals [26]. The lowest UPE-to-noise ratio was 4.0 ± 0.8 at pH 6.2. In the case of the system generating O_2_ (1Δg) the UPE increased only by 1.3- and 1.4 times at pH 6.6 and 6.8 in comparison to noise (PB with the addition of NaCl). Moreover, the ratio of UPE of Fe^2+^-H_2_O_2_ system to noise (which depends on the formation of O_2_ (1Δg) did not exceed 1.3 within the pH range from 6.0 to 7.4. Therefore, the contribution of 634 nm photons from decay of O_2_ (1Δg) to maximal UPE is low and could not be responsible for bias during estimation of anti-•OH activity of tested compounds using the Fe^2+^-EGTA-H_2_O_2_ system and light measurement with luminometer AutoLumat Plus LB 953. NaN_3_ is a frequently used scavenger of O_2_ (1Δg) especially generated by H_2_O_2_-NaClO system because it does not react with H_2_O_2_ and NaOCl [27]. NaN_3_ also quenched O_2_ (1Δg) dependent light emission from acetonitrile- H_2_O_2_ in alkaline environment [28]. We used a concentration of NaN_3_ comparable to that applied in the afore-mentioned experiments and therefore it is difficult to explain no inhibitory effect of this compound on the chemiluminescence of H_2_O_2_-NaOCl system. Thermal decomposition of NaN_3_ is accompanied by photon emanation [29] and may contribute to increased light emission from H_2_O_2_-NaN_3_-NaOCl samples. On the other hand, UPE of H_2_O-NaN_3_-NaCl did not differ from that of H_2_O-NaCl or NaCl alone making this explanation unlikely.

### 3.2. Effect of pH of Phosphate Buffer on Light Emission from the Fe^2+^-EGTA-H_2_O_2_ System

There are two reactions responsible for the generation of •OH radicals in the Fe^2+^-EGTA-H_2_O_2_ system.
Fe^2+^-EGTA + H_2_O_2_ → Fe^3+^-EGTA + OH^−^ + •OH(1)
Fe^3+^-EGTA + O_2_^•^- → Fe^2+^-EGTA + O_2_(2)

Chemical reaction (2) leads to regeneration of Fe^2+^-EGTA complex that can enter reaction (1) to increase •OH generation, oxidative damage to EGTA and formation of excited carbonyl groups with subsequent photon emission [20].

Two other reactions can be also involved in Fe^2+^ regeneration [30].
H_2_O_2_ + Fe^3+^-EGTA → HO_2_^•^ + H^+^ + Fe^2+^-EGTA
HO_2_^•^ + Fe^3+^-EGTA → O_2_ + H^+^ + Fe^2+^-EGTA

The ratio of H_2_O_2_ to Fe^2+^ in our Fe^2+^-EGTA-H_2_O_2_ system was around 28. Thus, the availability of Fe^2+^ is the limiting factor for •OH formation. Therefore, the reduction of Fe^3+^-EGTA to Fe^2+^-EGTA described by reaction (2) has an important contribution to total •OH radicals generation and light emission. An increase in pH above 6.8 was accompanied by a lower UPE of the Fe^2+^-EGTA-H_2_O_2_ system. Such conditions facilitate the formation of insoluble Fe(OH)_3_. Although the concentration of EGTA was 2-fold higher than that of iron ions, Fe^3+^ could be grabbed from the complex with EGTA and precipitated as Fe(OH)_3_, thus limiting Fe^2+^ regeneration. Five ferric hydrolysis products (Fe^3+^, FeOH_2_^+^, FeO_2_^−^, FeO_2_H and FeO^+^) can contribute to the total content of Fe^3+^ in aqueous solution shoving the complexity of chemical reactions involving Fe^2+^ and Fe^3+^ ions. The aqueous solubility of these products decreases across the pH range from 5.0 to 8.0 [31,32]. Especially Fe^3+^ solubility decreases from 10^−8^ mol/L, 7 × 10^−9^ mol/L to about 5 × 10^−9^ mol/L at pH 6.0, 6.8 and 7.0, respectively [31]. In our Fe^2+^-EGTA-H_2_O_2_ system, the concentration of Fe^2+^ ions was 92.6 μmol/L (mostly chelated with EGTA) and the concentration of H_2_O_2_ was 28- times higher than that of Fe^2+^. Assuming carefully that after injection of H_2_O_2_ to Fe^2+^-EGTA, only 10% of Fe^2+^ ions would be oxidized to Fe^3+^ over 2 min observation, the concentration of Fe^3+^ (9.26 × 10^−6^ mol/L) can be substantially higher than its solubility at pH 6.8 and 7.0 and result in Fe^3+^ precipitation and decrease in UPE of Fe^2+^-EGTA-H_2_O_2_.

Moreover, the reaction rate (1) depends on pH with maximal intensity at pH around 3 [33]. This may additionally elucidate the suppression of UPE when pH increased from 7.0 to 7.4. On the other hand, mean signal suppression in neutral (decrease by 1.19 times at pH 7.0) and alkaline environments (decrease by 1.24 times at pH 7.2) was not so great, taking into consideration the observation that, under these conditions, high-valent oxoiron [13,34] species are the main product of Fenton reaction. This is probably due to the presence of phosphate buffer (phosphates), which can augment •OH radicals formation even in moderate alkaline solutions [13].

Surprisingly, lowering the pH of the reaction milieu from 6.6 to 6.0 did not increase the UPE of the Fe^2+^-EGTA-H_2_O_2_ system. Quite the contrary, light emission decreased and reached the lowest values at pH 6.2. It should be pointed out that three radicals •OH, O_2_-, and O_2_ (1Δg) are produced in Fenton’s system [33]. The reactions leading to the production of these radicals can compete with each other and increased generation of one radical may affect the intensity of the remaining two reactions [33]. Thus increased generation of •OH radicals in a more acidic environment (reaction 1) may suppress the production of O_2_-radicals involved in the regeneration of Fe^2+^ ions and finally suppress the total yield of •OH radicals. It should be pointed out that •OH radicals can react with phosphate [13] which is a potential limitation in the use of phosphate buffer to stabilize the pH of the Fenton system reaction milieu. On the other hand, the kinetics of phosphate reactions with •OH radicals is significantly lower than that of the vast majority of organic compounds [13] including other buffers such as Tris and Hepes. Moreover, phosphate buffer plays a major role in maintaining the acid-base balance inside cells [35]. The intracellular concentration of free phosphates (H_2_PO^4−^ and HPO^4−^) is within the range of 0.5 mmol/L to 5 mmol/L [27]. In addition, concentration of labile organophosphates (e.g., phosphocreatine) is up to 20 times higher [36]. What is more, high UPE-to-noise (mean 9.7) and ΔUPE-to noise (mean 8.7) ratios at pH 6.6 suggest that scavenging of •OH radicals by phosphates did not substantially suppress light emission from the Fe^2+^-EGTA-H_2_O_2_ system which could be used for evaluation of anti-•OH radicals activity of various organic compound. Previously we tested four increasing concentrations of the Fe^2+^-EGTA-H_2_O_2_ system as an emitter of light under a stable ratio of Fe^2+^ to EGTA to H_2_O_2_ molar concentrations [20]. The optimal UPE was found for 92.6 µmol/L Fe^2+^-185.2 µmol/L EGTA-2.6 mmol/L H_2_O_2_ system [20]. Moreover, concentrations of 92.6 µmol/L Fe^2+^ and 2.6 mmol/L H_2_O_2_ correspond to some extent to the values that occur in vivo. The plasma levels of H_2_O_2_ and Fe complexed with low molecular weight compounds, can reach 50 µmol/L and 10 µmol/L in certain diseases [37,38]. It is believed that H_2_O_2_ concentrations can be even higher in a close neighborhood of activated inflammatory cells such as polymorphonuclear leukocytes and macrophages [39]. Additionally, the subcellular iron concentration calculated per average neuron of a brain can reach around 0.6 mM/L [40].

## 4. Materials and Methods

### 4.1. Chemicals and Solutions

All chemicals were of analytical grade. Sodium phosphate monobasic monohydrate (NaH_2_PO_4_·H_2_O), sodium phosphate dibasic heptahydrate (Na_2_HPO_4_·7H_2_O), iron (II) sulfate heptahydrate (FeSO_4_·7H_2_O), hydrochloric acid (HCl), sodium hydroxide (NaOH), sodium azide (NaN_3_), sodium hypochlorite (NaOCl), sodium chloride (NaCl), and ethylene glycol-bis (β-aminoethyl ether)-*N*,*N*,*N*′,*N*′-tetraacetic acid (EGTA) were purchased from Sigma-Aldrich Chemicals (St. Louis, MO, USA). H_2_O_2_ 30% solution (*w*/*w*) was from Chempur (Piekary Slaskie, Poland). Sterile deionized pyrogen-free water (freshly prepared, resistance > 18 MW/cm, HPLC H_2_O Purification System, USF Elga, Buckinghamshire, UK) was used throughout the study. Working aqueous solutions of 5 mmol/L FeSO_4_, 20 mmol/L NaN_3_ and 28 mmol/L NaCl were prepared before the assay. Thirty-percent solution of H_2_O_2_ was diluted with water to a final concentration of 28 mmol/L (working H_2_O_2_ solution) and the concentration was confirmed by the measurement of the absorbance at 240 nm using a molar extinction coefficient of 43.6/mol cm [41]. The stock solution of EGTA (100 mmol/L) was prepared in 10 mmol/L phosphate buffer pH = 8.0 and stored at room temperature in the dark for no longer than 3 months. A working solution of 10 mmol/L EGTA was obtained by the dilution of EGTA stock solution with water.

### 4.2. Effect of pH of Reaction Milieu on the Light Emission by Fe^2+^-EGTA-H_2_O_2_ System

Chemical reactions of 92.6 μmol/L Fe^2+^-185.2 μmol/L EGTA-2.6 mmol/L H_2_O_2_ system leading to UPE were carried out in 10 mmol/L phosphate buffers with pH ranging from 6.0 to 7.4 (pH = 6.0, 6.2, 6.4, 6.6, 6.8, 7.0, 7.2, and 7.4). UPE was measured with a multitube luminometer (AutoLumat Plus LB 953, Berthold, Germany) equipped with a Peltier-cooled photon counter (spectral range from 380 to 630 nm) to ensure high sensitivity and low and stable background noise signal. An amount of 10 mmol/L phosphate buffers with aforementioned pH were prepared following the prescription of AAT Bioquest (https://www.aatbio.com/resources/buffer-preparations-and-recipes/phosphate-buffer-ph-5-8-to-7-4, accessed on 19 August 2024). For instance, 10 mmol/L phosphate buffer (PB) pH = 6.0 was prepared by addition of 36.704 mg of Na_2_HPO_4_·7H_2_O and 0.119 mg of NaH_2_PO_4_·H_2_O to 80 mL of water, pH was adjusted to 6.0 with HCl and then distilled water was added to the final volume of 100 mL. Other buffers were done in the same way and amounts of Na_2_HPO_4_·7H_2_O and NaH_2_PO_4_·H_2_O were taken from AAT Bioquest page and pH was adjusted to the desired value with HCl or NaOH solutions. Briefly, 20 μL of 10 mmol/L EGTA solution was added to the tube (Lumi Vial Tube, 5 mL, 12 × 75 mm, Berthold Technologies, Bad Wildbad, Germany) containing 940 μL of PB (pH = 6.0). Then, 20 μL of 5 mmol/L solution of FeSO_4_ was added, and after gentle mixing, the tube was placed in the luminometer chain and incubated for 10 min in the dark at 37 °C. Then, 100 μL of 28 mmol/L H_2_O_2_ solution was added by an automatic dispenser and the total light emission (expressed in RLU) was measured for 120 s. The final concentrations of FeSO_4_, EGTA, and H_2_O_2_ in the reaction mixture were 92.6 µmol/L, 185.2 µmol/L, and 2.6 mmol/L, respectively. For experiments with other buffers (pH = 6.2, 6.4, 6.6, 6.8, 7.0, 7.2, and 7.4), the procedure was the same. Controls included: incomplete system I (Fe^2+^-H_2_O_2_ in PB); incomplete system II (EGTA-H_2_O_2_ in PB); H_2_O_2_ alone in PB; Fe^2+^ and EGTA in PB without H_2_O_2_; and medium alone (Table 2).

### 4.3. Effect of H_2_O_2_–NaOCl System on the Luminescence Signal Recorded by Luminometer with Photomultiplier Spectrum from 380 nm to 630 nm

During chemical reactions, three radicals are generated in the Fenton system [33]. One of them is O_2_ (1Δg) that after decay may emit photons with three bands (634 nm, 703 nm, and 1270 nm). The band of 634 nm is close to the upper border of the luminometer spectrum and may affect •OH—dependent UPE signal giving false positive results in our experiments. However, production of O_2_ (1Δg), in Fenton systems is much less intensive than generation of •OH radicals [42,43,44]. Therefore, to study the effect of O_2_ (1Δg) on the chemiluminescence signal we choose H_2_O_2_-NaOCl system a very effective generator of O_2_ (1Δg) [45]. The pH of chemical reaction environments 6.6 and 6.8 was chosen based on the results of experiments on effect of pH of reaction milieu on the light emission by Fe^2+^-EGTA-H_2_O_2_ system. UPE and ΔUPE (UPE minus baseline) reached the highest values at pH 6.6 and 6.8. To estimate this plausible effect, 100 µL of 28 mmol/L H_2_O_2_ solution was added to the tube containing 880 μL of 10 mmol/L PB pH = 6.8 and after gentle mixing the tube was placed in the luminometer chain and incubated for 10 min in the dark at 37 °C. Then, 100 μL of aqueous solution of 28 mmol/L NaOCl was added using an automatic dispenser and the total light emission (expressed in RLU) was measured for 120 s. It should be pointed out that NaOCl solution was prepared in ice-cold water and kept in an ice bath throughout the whole experiment. The final concentrations of H_2_O_2_ and NaOCL were 2.6 mmol/L. The design of these experiments and control systems are shown in Table 3.

### 4.4. Statistical Analyses

Results obtained from 9 series of separate experiments are expressed as means (standard deviations) and medians and interquartile ranges (IQR) of relative light units (RLU). The following parameters were recorded and calculated: UPE (ultra-weak photons emission)—total light emission within the first two minutes after the addition of H_2_O_2_ or H_2_O and NaOCl or NaCl, increment in UPE (ΔUPE = UPE of a given system—UPE of buffer alone (noise), and the ratio of UPE (or ΔUPE) of a given system to noise when pH of reaction milieu increased from 6.0 to 7.4. The comparisons between the UPE and ΔUPE and their ratios to noise observed in different pH of reaction milieu were analyzed with the independent-samples (unpaired) t-test or Mann–Whitney U test depending on the data distribution, which was tested with the Kolmogorov–Smirnov–Liliefors test. The Brown–Forsythe test for analysis of the equality of the group variances was used prior to the application of the unpaired t-test and if variances were unequal, the Welch’s t-test was used instead of the standard *t*-test. A *p*-value < 0.05 was considered significant.

## 5. Conclusions

We found that •OH radicals-induced light emission from the Fe^2+^-EGTA-H_2_O_2_ system is highest at pH 6.6 stabilized with 10 mmol/L phosphate buffer. Both increase in pH within the range of 6.8 to 7.4 and decrease from 6.4 to 6.0, resulting in suppression of UPE and a decrease in the UPE-to-noise ratio. The following processes summarized in Figure 2 may be responsible for this phenomenon;

Enhancement and inhibition of •OH production in lower and higher pH, respectively.Formation of insoluble and non-reactive Fe(OH)_3_ at neutral and alkaline environment.Enhancement of •OH production by phosphates at weakly acidic and neutral environments.Suppression of O_2_^•^-production in acidic environment with decreased intensity of Fe^2+^-EGTA complex regeneration.

Phosphates are the main intracellular buffer and the pH range investigated in our study occurs in human body cells. Therefore, the Fe^2+^-EGTA-H_2_O_2_ system with pH 6.6 stabilized with PB resembles intracellular conditions and seems optimal for the determination of anti-•OH activity of the variety of water-soluble organic compounds.

## Figures and Tables

**Figure 1 molecules-29-04014-f001:**
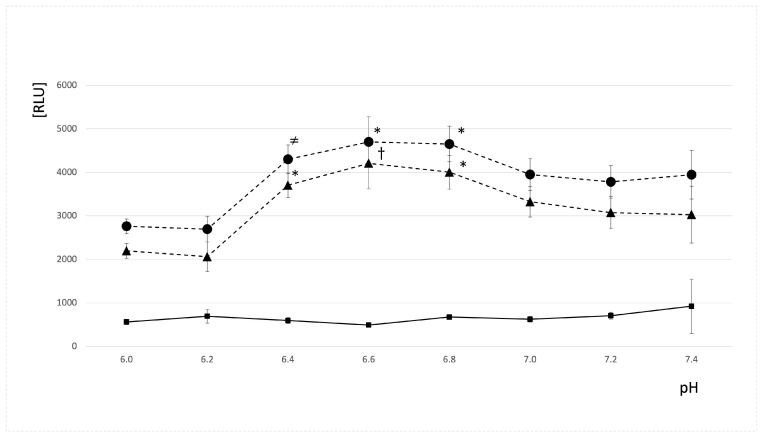
Effect of pH of reaction milieu on ratios of the UPE of the Fe^2+^-EGTA-H_2_O_2_ system to noise (-●-), ΔUPE of the Fe^2+^-EGTA-H_2_O_2_ system to noise (-▲-), and the UPE of the Fe^2+^-H_2_O_2_ system to noise (-■-). Each point represents the mean ± SD of nine series of separate experiments. *—significantly different from all corresponding values—*p* < 0.05. †—significantly different from corresponding values noted for pH = 6.0, 6.2, 6.6, 7.0, 7.2, and 7.4—*p* < 0.05. ≠—significantly different from corresponding values noted for pH = 6.0, 6.2, 6.6, 7.2, and 7.4—*p* < 0.05. Each point represents the mean ± SD of nine series of separate experiments.

**Figure 2 molecules-29-04014-f002:**
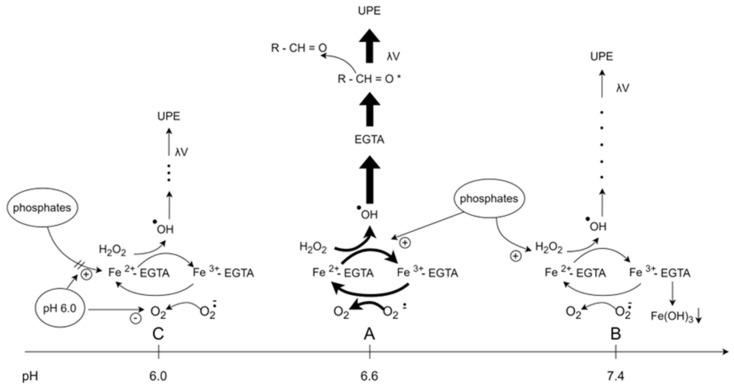
The proposed mechanisms for the effect of pH changes from 6.0 to 7.4 of reaction milieu on UPE of 92.6 μmol/L Fe^2+^-185.2 μmol/L EGTA-2.6 mmol/L H_2_O_2_ system. The pH range from 6.0 to 7.4 was studied. (**A**)—Under conditions of pH = 6.6 the UPE (ultra weak photon emission) was maximal. Hydroxyl radicals (•OH) generated in the reaction of Fe^2+^-EGTA with H_2_O_2_ (Fenton reaction) attack one of the ether bonds in the backbone structure of EGTA resulting in the formation of product with triplet excited carbonyl group (R-CH = O*). Electronic transitions from the triplet excited state to the ground state is accompanied by the photon emission (λν). Superoxide radicals (O_2_^.−^) produced simultaneously in the Fenton system reduce Fe^3+^-EGTA to Fe^2+^-EGTA that again enters the Fenton reaction increasing a number of emitted photons. Additionally, phosphate anions (H_2_PO^4−^/HPO4^2−^) augment the intensity of the Fenton reaction. (**B**)—When pH increased from 6.6 to 7.4, the rate of the Fenton reaction decreased and part of Fe^3+^ formed insoluble Fe(OH)_3_ and less Fe^2+^-; EGTA is available for reaction with H_2_O_2_. Although the rate of Fenton reaction is still stimulated by H_2_PO^4−^/HPO_4_^2−^, the net formation of •OH is decreased and UPE lowered. (**C**)—A decrease in pH from 6.6 to 6.0 resulted in a moderate increase in the rate of the Fenton reaction, while the stimulatory effect of H_2_PO4^−^/HPO4^2−^ phosphate anions was abolished. Parallel production of O_2_- is decreased and in consequence, regeneration of Fe^2+^-EGTA diminished. The net yield of •OH production is decreased and less R-CH = O* is formed with subsequent emission of photons. Thus, the UPE is lower than that observed under the condition of pH = 6.6.

**Table 1 molecules-29-04014-t001:** Singlet oxygen (O_2_ (1Δg))-dependent chemiluminescence signal [RLU] recorded by a luminometer with a photomultiplier spectrum from 380 nm to 630 nm.

No.	Sample	pH of Reaction Milieu (10 mmol/L PB) pH = 6.8	pH of Reaction Milieu (10 mmol/L PB) pH = 6.6
1.	Complete system H_2_O_2_-NaOCl	926 ± 245 (878; 218) *	809 ± 124 (827; 105) *^,^†
2.	Complete system with O_2_ (^1^Δg) scavenger-H_2_O_2_-NaN_3_-NaOCl	1073 ± 105 (1102; 96)	930 ± 93 (974; 161)
3.	Incomplete system I H_2_O_2_-NaCl	628 ± 74 (651; 120)	641 ± 78 (653; 129)
4.	Incomplete system II H_2_O-NaOCl	651 ± 80 (695; 133)	654 ± 85 (669; 131)
5.	H_2_O-NaN_3_-NaCl	640 ± 46 (648; 50)	636 ± 54 (643; 84)
6.	H_2_O-NaCl	609 ± 60 (630; 107)	661 ± 104 (659; 101)
7.	NaCl	640 ± 47 (649; 65)	635 ± 56 (653; 121)

Total light emission was measured for 2 min just after the automatic injection of 100 μL of NaOCl or NaCl solution. Final sample volume 1080 μL. Results are expressed as mean and standard deviation and (median; interquartile range) of RLU. The final concentrations of H_2_O_2_, NaOCl, NaCL, and NaN_3_ were 2.6 mmol/L, 2.6 mmol/L, 2.6 mmol/L, and 18.5 mmol/L. *—vs. corresponding values obtained for samples No 3, 4, 5, 6, and 7—*p* < 0.05. †—vs. corresponding value for sample No 2—*p* < 0.05.

**Table 2 molecules-29-04014-t002:** Design of experiments on the effect of pH of reaction milieu on light emissions by the Fe^2+^-EGTA-H_2_O_2_ system.

No.	Sample	Working Solutions Added to Luminometer Tube (µL)				
		A-PB	B-EGTA	C-FeSO_4_	D-H_2_O	E-H_2_O
1.	Complete system	940	20	20	100	0
2.	Incomplete system I	960	0	20	100	0
3.	Incomplete system II	960	20	0	100	0
4.	H_2_O_2_ alone	980	0	0	100	0
5.	Fe^2+^ + EGTA without H_2_O_2_	940	20	20	0	100
6.	Medium alone	980	0	0	0	100

Working solutions were mixed in alphabetical order. (A)—10 mM phosphate buffer (PB) with increasing pH from 6.0 to 7.4 (pH = 6.0, 6.2, 6.4, 6.6, 6.8, 7.0, 7.2, and 7.4); (B)—10 mmol/L aqueous solution of ethylene glycol-bis (β-aminoethyl ether)-*N*,*N*,*N*′,*N*′-tetraacetic acid (EGTA), and (C)—5 mmol/L solution of FeSO_4_. Then, after gentle mixing the tube was placed into a luminometer chain, incubated for 10 min at 37 °C and then (D)—28 mmol/L aqueous solution of H_2_O_2_ or (E)—H_2_O was automatically injected with dispenser and total light emission was measured for 2 min.

**Table 3 molecules-29-04014-t003:** Design of experiments on the effect of singlet oxygen (O2 (1Δg)) generating system (H_2_O_2_-NaOCl) on the luminescence signal recorded by luminometer with photomultiplier spectrum from 380 nm to 630 nm.

No.	Sample	Volumes of Working Solutions Added to Luminometer Tube (μL)					
		A-PB	B-H_2_O_2_	C-H_2_O	D-NaN3	E-NaOCl	F-NaCl
1.	Complete system	880	100	0	0	100	0
2.	Incomplete system I	880	0	100	0	100	0
3.	Complete system + NaN_3_	780	100	0	100	100	0
Additional control							
4.	Incomplete system II	880	100	0	0	0	100
5.	Medium alone	880	0	100	0	0	100
6.	Medium + NaN_3_	780	0	100	100	0	100
7.	PB alone	980	0	0	0	0	100

Working solutions were mixed in alphabetical order. (A)—10 mM phosphate buffer (PB) (pH = 6.8); (B)—28 mmol/L solution of H_2_O_2_, (C)—distilled water, (D)—20 mmol/L NaN_3_—an O2 (1Δg) scavenger, then after gentle mixing, the tube was placed into a luminometer chain, incubated for 10 min at 37 °C, and then (E)—28 mmol/L aqueous solution of NaOCl or (F)—28 mmol/L NaCl (additional controls) was automatically injected with dispenser and total light emission was measured for 2 min.

## Data Availability

The data presented in this study are available on request from the corresponding author.

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
