# Peer review of "Light Emission from Fe2+-EGTA-H2O2 System Depends on the pH of the Reaction Milieu within the Range That May Occur in Cells of the Human Body"

_molecules, 2024, doi:10.3390/molecules29174014_

Round 1

Reviewer 1 Report

Comments and Suggestions for Authors

The main question addressed by the research shows that the generation of •OH and light emission due to oxidative damage to EGTA from Fe2+-EGTA-H2O2 system may depend on the buffer pH of the reaction milieu. To answer this question, the author evaluated the effect of pH from 6.0 to 7.4 stabilized with 10 mM phosphate buffer on chemiluminescence signal and the ratio of this signal to noise (light emission from medium alone).

 Comments and questions:

1.      The conclusion is that formation of insoluble Fe(OH)3 at neutral and alkaline environments cause a pH above 6.8 with a lower UPE signal is not consistent with the evidence. Because it’s a hypothesis, unless provide evidence of Fe(OH)3 formation.

2.      Briefly introduce why you should study the section 2.1 “Effect of singlet oxygen (O2 (1Δg)) generating system (H2O2-NaOCl) on the luminescence signal recorded,” and why choose pH 6.8 and pH6.6 in this part?

3.       Insert samples No in table 1, make samples No 3, 4, 5, 6, and 7 in line 102 more clear.

4.        Figure 2 Effect of pH of reaction milieu on ratios of UPE is more representative that Figure 1, there is no need to put Figure 1 in the main text.

5.      The concentration of Fe2+, EGTA, and H2O2 should be important in this system, which needs to be optimized.

6.       Test Fe2+-EGTA-H2O2 system in other commonly used intracellular mimicking buffers.  

In summary, I recommend this paper be majorly revised before accepting.

Author Response

  1. The conclusion is that the formation of insoluble Fe(OH)3 at neutral and alkaline environments causes a pH above 6.8 with a lower UPE signal is not consistent with the evidence because it’s a hypothesis unless providing evidence of Fe(OH)3

Response:  In discussion, we added the following fragment that describes the formation and precipitation of Fe3+ ions: “Five ferric hydrolysis products (Fe3+, FeOH2+, FeO2-, FeO2H and FeO+) can contribute to the total content of Fe3+ in aqueous solution shoving the complexity of chemical reactions involving Fe2+ and Fe3+ ions. The aqueous solubility of these products decreases across the pH range from 5.0 to 8.0 [31,32]. Especially Fe3+ solubility decreases from 10-8 mol/L, 7x10-9 mol/L to about 5x10-9 mol/l at pH 6.0, 6.8 and 7.0, respectively [31]. In our Fe2+ - EGTA - H2O2 system the concentration of Fe2+ ions was 92.6 μmol/L (mostly chelated with EGTA) and the concentration of H2O2 was 28- times higher than that of Fe2+. Assuming carefully that after injection of H2O2 to Fe2+-EGTA only 10% of Fe2+ ions would be oxidized to Fe3+ over 2 min observation, the concentration of Fe3+ (9.26 x 10-6 mol/L) can be substantially higher than its solubility at pH 6.8 and 7.0 and result in Fe3+ precipitation and decrease in UPE of Fe2+-EGTA -H2O2

  1. Briefly introduce why you should study section 2.1 “Effect of singlet oxygen (O2 (1Δg)) generating system (H2O2-NaOCl) on the luminescence signal recorded,” and why choose pH 6.8 and pH6.6 in this part?

Response: We added the following fragment explaining why we studied the effect of singlet oxygen (O2 (1Δg)) generating system (H2O2-NaOCl) on the luminescence signal recorded and why we chose reaction milieu with pH 6.8 and pH 6.6    “ During chemical reactions, three radicals are generated in the Fenton system [33]. One of them is O2 (1Δg) which after decay may emit photons with three bands (634 nm, 703 nm, and 1270 nm). The band of 634 nm is close to the upper border of the luminometer spectrum and may affect •OH – dependent UPE signal giving false positive results in our experiments. However, production of O2 (1Δg), in Fenton systems is much less intensive than generation of •OH radicals [42-44]. Therefore, to study the effect of O2 (1Δg) on the chemiluminescence signal we choose H2O2 -NaOCl system a very effective generator of O2 (1Δg) [45]. The pH of chemical reaction environments 6.6 and 6.8 was chosen based on the results of experiments on the effect of the pH of the reaction milieu on the light emission by the Fe2+ – EGTA – H2O2 system. UPE and ΔUPE (UPE minus baseline) reached the highest values at pH 6.6 and 6.8.; “

  1. Insert sample No in Table 1, and make samples No 3, 4, 5, 6, and 7 in line 102 more clear.

Response: Table 1 (as well as other Tables) has been modified according to the Reviewer’s suggestion

  1. Figure 2 Effect of pH of reaction milieu on ratios of UPE is more representative than Figure 1, there is no need to put Figure 1 in the main text.

Response: According to the reviewer’s suggestion, Figure 2 is in the main text while Figure 1 was implemented as supplementary material. Thus, there are 3 figures in the main text and one in the supplementary material

5.      The concentration of Fe2+, EGTA, and H2O2 should be important in this system, which needs to be optimized.

Response: We briefly described our previous experiments on the optimization of the Fe2+ - EGTA – H2O2 system as well as in vivo factors determining the selection of Fe2+ and H2O2 concentrations that are close to those occurring in extracellular and intracellular fluid. This fragment is as follows; “Previously we tested four increasing concentrations of Fe2+- EGTA- H2O2 system as an emitter of light under a stable ratio of Fe2+ to EGTA to H2O2 molar concentrations [20]. The optimal UPE was found for 92.6 µmol/L Fe2+-185.2 µmol/L EGTA-2.6 mmol/L H2O2 system [20]. Moreover, concentrations of 92.6 µmol/L Fe2+ and 2.6 mmol/L H2O2 correspond to some extent to the values that occur in vivo. The plasma levels of H2O2 and Fe complexed with low molecular weight compounds, can reach 50 µmol/L and 10 µmol/L in certain diseases [37,38]. It is believed that H2O2 concentrations can be even higher in a close neighborhood of activated inflammatory cells such as polymorphonuclear leukocytes and macrophages [39]. Additionally, the subcellular iron concentration calculated per average neuron of a brain can reach around 0.6 mM/L [40].”

  1. Test the Fe2+-EGTA-H2O2 system in other commonly used intracellular mimicking buffers.  

Response:  We listed other intracellular buffers and gave the data about their reactivity with hydroxyl radicals and plausible consequences for light emission (inhibition or enhancement) from  Fe2+ - EGTA – H2O2 system. This fragment is as follows:”. It should be pointed out that Fenton’s systems dedicated to the evaluation of anti-•OH activity must resemble in vivo conditions and be relatively simple to facilitate the interpretation of obtained results. Bicarbonate and phosphate buffers are the main buffers in extracellular (e.g. blood) and intracellular fluid [14], respectively. There are also other intracellular buffers such as amino-acids, proteins, and organic acids (mainly carboxylic acids) (e.g. acetic acid, lactic acid, citric acid, succinic acid) that with their dissociated form can stabilize pH inside the cell [14]. On the other hand, •OH radicals can effectively oxidize the aforementioned intracellular buffers [15,16]. Moreover, oxidation of some proteins and tryptophane can lead to the formation of excited carbonyl groups with subsequent photon emission [17-19] making results obtained with the Fe2+-EGTA-H2O2 system almost impossible to interpret. Therefore, phosphate buffer composed of dibasic sodium phosphate and monobasic sodium phosphate seems the optimal intracellular buffer that can be used for investigation of the effect of pH changes on UPE of Fe2+ - EGTA- H2O2 system.”  

Reviewer 2 Report

Comments and Suggestions for Authors

I suggest a major revision for the entitled manuscript Light emission from Fe2+-EGTA-H2O2 system depends on the pH of the reaction milieu within the range that may occur in cells of the human body. My specific comments are as follows:

1. Remove the full stop from the title.

2. Introduction should be separated to several paragraphs. And the novelty of this work should be proposed clearly in introduction.

3. About the formation of insoluble and non-reactive Fe(OH)3, was there any result and figure?

4. Section 3.2, was there any experiment to directly prove the result about the cycle of Fe3+ to Fe2+? The authors can improve the discussion about conversion of Fe3+ to Fe2+, please refer the paper (10.1016/j.cclet.2023.109185) and add to reference.

5. The reason for enhancement and inhibition of •OH production in lower and higher pH should be in-depth explanation and discussion.

Author Response

I suggest a major revision for the entitled manuscript Light emission from Fe2+-EGTA-H2O2 system depends on the pH of the reaction milieu within the range that may occur in cells of the human body. My specific comments are as follows:

  1. Remove the full stop from the title.

Response: We removed the full stop from the title.

  1. The introduction should be separated into several paragraphs. And the novelty of this work should be proposed clearly in the introduction. Response: According to the Reviewer’s suggestion we separated the Introduction into three paragraphs :

1.1. Fenton systems as a tool for determination of anti- (•OH) activity and limiting buffer interactions 

1.2. Properties of Fe2+ - EGTA – H2O2 system generating ultra-weak chemiluminescence

1.3. Aims of the study

The first part of the introduction was increased by the addition of the following text: . It should be pointed out that Fenton’s systems dedicated to the evaluation of anti- •OH activity must resemble in vivo conditions and be relatively simple to facilitate the interpretation of obtained results. Bicarbonate and phosphate buffers are the main buffers in extracellular (e.g. blood) and intracellular fluid [14], respectively. There are also other intracellular buffers such as amino-acids, proteins, and organic acids (mainly carboxylic acids) (e.g. acetic acid, lactic acid, citric acid, succinic acid) that with their dissociated form can stabilize pH inside the cell [14]. On the other hand, •OH radicals can effectively oxidize the aforementioned intracellular buffers [15,16]. Moreover, oxidation of some proteins and tryptophane can lead to the formation of excited carbonyl groups with subsequent photon emission [17-19] making results obtained with the Fe2+-EGTA-H2O2 system almost impossible to interpret. Therefore, phosphate buffer composed of dibasic sodium phosphate and monobasic sodium phosphate seems the optimal intracellular buffer that can be used for investigation of the effect of pH changes on UPE of Fe2+ - EGTA- H2O2 system.”

The third part of the introduction was increased by the addition of the following text: We found that decay of O2 (1Δg) did not significantly affect of UPE of Fe2+ - EGTA – H2O2 system and pH = 6.6 of reaction milieu results in maximal signal-to-noise ratio being optimal for in vitro determination of anti-•OH activities of various compounds.   

  1. About the formation of insoluble and non-reactive Fe(OH)3, was there any result and figure?

Response: This topic is discussed (Discussion section) and is as follows: “Five ferric hydrolysis products (Fe3+,FeOH2+ , FeO2- , FeO2H and FeO+) can contribute to the total content of Fe3+ in aqueous solution shoving the complexity of chemical reactions involving Fe2+ and Fe3+ ions. The aqueous solubility of these products decreases across the pH range from 5.0 to 8.0 [31,32]. Especially Fe3+ solubility decreases from 10-8 mol/L, 7x10-9 mol/L to about 5x10-9 mol/l at pH 6.0, 6.8 and 7.0, respectively [31]. In our Fe2+ - EGTA - H2O2 system the concentration of Fe2+ ions was 92.6 μmol/L (mostly chelated with EGTA) and the concentration of H2O2 was 28- times higher than that of Fe2+. Assuming carefully that after injection of H2O2 to Fe2+-EGTA only 10% of Fe2+ ions would be oxidized to Fe3+ over 2 min observation, the concentration of Fe3+ (9.26 x 10-6 mol/L) can be substantially higher than its solubility at pH 6.8 and 7.0 and result in Fe3+ precipitation and decrease in UPE of Fe2+-EGTA -H2O2

  1. Section 3.2, was there any experiment to directly prove the result about the cycle of Fe3+ to Fe2+? The authors can improve the discussion about the conversion of Fe3+ to Fe2+, please refer to the paper (10.1016/j.cclet.2023.109185) and add to the reference.

Response: Two additional chemical reactions (based on the afore-mentioned paper) leading to the regeneration of Fe2+ are described in the modified Discussion:” Two other reactions can be also involved in Fe2+ regeneration [30]

H2O2 + Fe3+ - EGTA → HO2 + H+ + Fe2+- EGTA

HO2 + Fe3+ - EGTA →  O2 + H+ + Fe2+-EGTA

  1. The reason for enhancement and inhibition of •OH production in lower and higher pH should be an in-depth explanation and discussion.

Response:  please refer to point 3 describing precipitation of Fe3+  at pH higher than 6.6 and thus limiting regeneration of Fe2+

Due to improvements in the Introduction, Material and Methods, and Discussion the number of cited references increased by 14.

Round 2

Reviewer 1 Report

Comments and Suggestions for Authors

Accept

Reviewer 2 Report

Comments and Suggestions for Authors

It has been improved, can be accepted.